# Transgene-Free Genome Editing for Biotic and Abiotic Stress Resistance in Sugarcane: Prospects and Challenges

Sakthivel Surya Krishna [1], S R Harish Chandar [1], Maruthachalam Ravi [2], Ramanathan Valarmathi [1], Kasirajan Lakshmi [1], Perumal Thirugnanasambandam Prathima [1], Ramaswamy Manimekalai [1], Rasappa Viswanathan [3], Govindkurup Hemaprabha [1] and Chinnaswamy Appunu [1,*]

[1]  Division of Crop Improvement, Indian Council of Agricultural Research-Sugarcane Breeding Institute, Coimbatore 641007, Tamil Nadu, India
[2]  Indian Institute of Science Education and Research (IISER), Thiruvananthapuram 695551, Kerala, India
[3]  ICAR—Indian Institute of Sugarcane Research, Lucknow 226002, Uttar Pradesh, India
*  Correspondence: cappunu@gmail.com or C.Appunu@icar.gov.in

**Abstract:** Sugarcane (*Saccharum* spp.) is one of the most valuable food and industrial crops. Its production is constrained due to major biotic (fungi, bacteria, viruses and insect pests) and abiotic (drought, salt, cold/heat, water logging and heavy metals) stresses. The ever-increasing demand for sugar and biofuel and the rise of new pest and disease variants call for the use of innovative technologies to speed up the sugarcane genetic improvement process. Developing new cultivars through conventional breeding techniques requires much time and resources. The advent of CRISPR/Cas genome editing technology enables the creation of new cultivars with improved resistance/tolerance to various biotic and abiotic stresses. The presence of genome editing cassette inside the genome of genome-edited plants hinders commercial exploitation due to regulatory issues. However, this limitation can be overcome by using transgene-free genome editing techniques. Transgene-free genome editing approaches, such as delivery of the RNPs through biolistics or protoplast fusion, virus-induced genome editing (VIGE), transient expression of CRISPR/Cas reagents through *Agrobacterium*-mediated transformation and other approaches, are discussed. A well-established PCR-based assay and advanced screening systems such as visual marker system and Transgene killer CRISPR system (TKC) rapidly identify transgene-free genome edits. These advancements in CRISPR/Cas technology speed up the creation of genome-edited climate-smart cultivars that combat various biotic and abiotic stresses and produce good yields under ever-changing conditions.

**Keywords:** sugarcane; transgene-free genome editing; biotic stress; abiotic stress

## 1. Introduction

Sugarcane (*Saccharum* spp.) is one of the most important food and industrial crops cultivated across the world [1]. Sugarcane contributes to 80% of global sugar production with India as the leading sugar producer followed by Brazil, European Union, Thailand and China [2]. In addition to this, sugarcane juice and the by-product, molasses, are the major raw materials used in distilleries for alcohol and bioethanol production [1]. The crushed sugarcane stalks are burned to generate electricity. Thus, sugarcane plays an indispensable role in supplying raw materials for the food and fuel industries. The human population reached approximately 7.95 billion in 2022 and it is projected that the global population will reach 9.7 billion in 2050 [3]. The pollution caused by industries, urban sewage and, heavy use of chemical inputs in agricultural fields causes severe damage to land and water resources, which in turn have serious impacts on all-life forms on Earth. The lethal impacts of climate change on agriculture are the epidemic outbreak of new pests and diseases and the periodic occurrence of drought and waterlogging. Sugarcane is adversely affected by various biotic and abiotic stresses. Biotic stresses, such as insect pests, diseases and

noxious weeds, severely affect cane production and juice quality [4–6]. Severe drought due to monsoon failure, excessive rainfall, low temperature, salinity, alkalinity and heavy metal contamination in soil increase day-by-day and also have an adverse effect on cane production and juice quality [7–10].

Conventional breeding and molecular marker-assisted selection are carried out in sugarcane to develop new cultivars. Transgenic events were also developed in sugarcane for pest and disease resistance, but their commercialization has not been successful due to various regulatory issues [11]. Although conventional breeding strategies have led to commercial success, more time is required to develop cultivars. The polyploid nature of sugarcane makes crop improvement programs much more complicated than for diploid plants. The increasing human population and adverse impacts of climate change on plants and animals call for the use of sustainable modern techniques in crop production to meet global food demands and to ensure a better way of living. The advent of genome editing technologies, such as zinc finger nucleases (ZFNs), transcription activator-like effector nucleases (TALENS), clustered regularly interspaced short palindromic repeats and CRISPR-associated protein 9 (CRISPR/Cas9), expedites the crop improvement process by creating site-specific mutations in the plant genome [12]. Among these, CRISPR/Cas9 is widely used because of its high editing efficiency, cost-effectiveness, and multiplexing [13].

CRISPR/cas9 consists of two main components: a guide RNA and Cas9 endonuclease. The single-stranded guide RNA consists of a 20 bp sequence complementary to the host target region, and Cas9 induces DSBs (Double-stranded breaks) in the 3 bp upstream of PAM sequence (5′NGG3′) in the DNA of the host, complementary to the guide RNA (gRNA) [14,15]. The host cell DNA repair machinery repairs the breaks by either NHEJ (Non-homologous end joining) or HDR (Homology directed repair). NHEJ is predominant and error-prone, thus creating mutations by base insertions, deletions and replacements [16–18]. Currently, many Cas variants other than Cas9 have been identified and used for genome editing purposes. The problem with using vector-based delivery of the CRISPR/Cas system is that the markers, promoters and Cas9 genes in the genome editing cassette integrate into the plant genome and are passed on to successive generations, which places the genome-edited (GE) plants developed through this technique under the category of genetically modified organisms (GMOs). In the case of seed propagated annuals, the transgenes can be eliminated by recombination and selection. Sugarcane is a highly heterozygous vegetatively propagated crop; eliminating the transgenes via meiotic recombination and producing GE lines with exact genome composition other than the difference in target region is highly difficult. However, the advent of transgene-free genome-editing techniques enables the creation of site specific mutations without integration of any residual transgenes in the plant genome and these techniques are very useful in creating GE sugarcane without any residual foreign genes. The objective of this review is to highlight strategies to develop and screen transgene-free GE sugarcane lines for rapid genetic improvement with special emphasis on imparting tolerance/resistance to major biotic and abiotic stresses.

## 2. Biotic Stresses Affecting Sugarcane

Diseases and insect pests adversely affect sugarcane and incur a huge yield loss, increasing the cost of cultivation [4]. As sugarcane is allowed to produce ratoons, the complete elimination of established diseases and insect pests is an arduous task to accomplish. However, the extent of damage caused by biotic agents can be reduced by adapting various agronomic practices.

### 2.1. Major Diseases

The major fungal diseases of sugarcane are red rot, wilt and smut [4]. Red rot caused by *Colletotrichum falcatum* is one of sugarcane's most devastating diseases that severely affects juice quality by disrupting the sucrose metabolism [4]. The affected canes have a reddish discoloration in the stalk portion [4]. Wilt is caused by *Fusarium sacchari*, which causes yield

loss of up to 15% and reduces sugar recovery during extraction [19]. The symptoms are yellowing and drying of canes with pinkish-brown discoloration in the internal cavities [19]. Smut is caused by *Sporisorium scitamineum* which causes yield losses of up to 50% and reduces sugar recovery. The symptoms of smut are affected cane shoots modified into sorus or a whip-like structures [20]. The other fungal diseases of sugarcane are pineapple disease, rust, pokkah boeng, stalk rot, brown spot, eye spot, ring spot, seedling rot and many others [4].

The major bacterial diseases affecting sugarcane are ratoon stunting and leaf scald. Ratoon stunting is caused by *Leifsonia (Clavibacter) xyli subsp. xyli* causes yield loss up to 30% [21]. The symptoms of the disease are stunted growth with thin stalks, reduction in tillering, shortened internodes and yellowing of the leaf. Leaf scald is caused by *Xanthomonas albilineans*; the pathogen shows acute and chronic expression [22]. During the chronic phase, the pathogen produces white lines along the leaf blade to the leaf sheath, followed by chlorosis, necrosis and death of young shoots and stalks. The acute phase is initially symptomless followed by the sudden outbreak and death of plants [22,23].

Sugarcane grassy shoot (SCGS) disease is a phytoplasma disease of sugarcane transmitted by leaf hoppers which causes huge yield loss in susceptible varieties. The symptoms of SCGS are excessive tillering with thin slender tillers and shortened internodes showing grass-like appearance with chlorosis. The affected canes fail to produce millable canes [4,24]. Yellow leaf is one of the most destructive diseases caused by the sugarcane yellow leaf virus (ScYLV), which accounts for 15–50% yield loss of sugarcane [25–28]. The symptoms of yellow leaf disease appear on 6–8-month-old crops and the symptoms are prominent midrib yellowing on the lower surface of the leaf followed by the lateral spread of yellowing and leaf tip necrosis, shortening of internodes and bunchy top appearance of leaves during severe infection [25–28]. Sugarcane mosaic disease is caused by three viruses, namely sugarcane mosaic virus (SCMV), sugarcane streak mosaic virus (SCSMV) and sorghum mosaic virus (SrMV), which cause severe yield losses of more than 50% in case of severe infection. The symptoms of the mosaic disease are yellow and green spots on the young leaves with interveinal chlorosis; upon severe infection, leaves turn yellow or yellow-white with green or necrotic patches [29–31].

*2.2. Major Insect Pests*

Pest infestation causes severe yield losses in sugarcane. The major insect pests affecting sugarcane are early shoot borer, top borer, internode borer, termites, root grub, aphids and white fly. The early shoot borer (*Chilo infuscatellus*) affects canes during the early crop stages from one to three months; the symptoms are bore holes in the shoot above ground level, dead heart and rotten portions that emit an offensive odour, and affected canes that can be easily pulled out [5,32,33]. The internode borer (*Chilo sacchariphagus indicus*) affects the crop soon after internodes emerge; the boreholes are plugged with excreta and frass is seen in the infected portions. The internodes are reduced in length and girth [5,33,34]. The top shoot borer (*Scirpophaga excerptalis*) affects the crop during all growth stages; its characteristic symptoms are bore holes in the apical portion of the stem, shot holes on opened leaves, dead heart in the fully grown canes and bunchy top formation [5,33,35]. The borers account for yield losses of 15–42% [5,33,36].

Whitefly (*Aleurolobus barodensis* and *Neomaskellia andropogonis*) is a serious sucking sugarcane pest that causes yield reduction up to 50% [37,38]. The symptoms are yellowing of leaves followed by drying and retarded growth [37,38]. The sugarcane woolly aphid (*Ceratovacuna lanigera*) produces symptoms similar to that of whitefly. Honeydew secretions from whitefly and aphids favour the growth of sooty mould fungus [33].

Subterranean pests, such as termites and white grubs, are difficult to control and cause severe yield loss. White grub (*Holotrichia serrata*) is another serious sugarcane pest that can cause yield losses up to 100% [33,39]. The larvae feed voraciously on the root system, leading to poor growth, lodging, yellowing and wilting of canes. The affected canes can be easily pulled out and the damage usually occurs in patches in the field [33,39].



Termites (*Odontotermes obesus*) mainly affect the germinating setts in the field, causing poor germination. The damaged portions are filled with soil, and damage in the later stages of the crop causes yellowing and drying of leaves [5,33].

## 3. Abiotic Stress in Sugarcane

Worldwide sugarcane production is affected by abiotic stresses, namely drought, salinity, cold and heat stress, and metal toxicity. These stresses incur huge economic loss to farmers and severely restrict crop cultivation in stress-prone areas.

### 3.1. Drought

Water deficit caused by meteorological and agricultural drought leads to drought stress [40]. Lack of water during key water requirement periods, such as germination, tillering, grand growth phase and maturation, reduces cane development and affects the sugar content [41]. Yield loss due to lack of water in sugarcane is estimated to be approximately 50–70% based on the degree of water stress [42]. Plants use various morphological and physiological strategies to mitigate drought stress, which varies from genotype to genotype [43]. The plant responds to water stress by exhibiting physiological changes such as leaf rolling, stomatal closure, inhibition of stalk and leaf growth, leaf senescence and reduced leaf area [44]. Water stress reduces biochemical activities in mesophyll and bundle sheath cells which in turn reduces the sugar accumulation in the plant [45]. The accumulation of ROS, such as superoxide radicals ($O_2^-$), hydrogen peroxide ($H_2O_2$) and hydrogen radicals ($OH\cdot$), in the cells causes severe damage to cell components such as chlorophyll, lipids, proteins and nucleic acids [46,47]. The ROS also acts as a secondary messenger in response to stress conditions. Hormones play an important role as messengers that regulate plant's responses to lack of soil water. Abscisic acid (ABA) synthesized by plant roots, which control the stomatal opening and closure, produces the first signal during plant stress and is transported through the xylem to the roots [48]. Controlled and regulated opening and closing of stomata is required to avoid loss of water through transpiration during water deficit conditions [49].

### 3.2. Salt

Since sugarcane is a glycophyte, it is highly sensitive to salinity [50]. Germination and early growth stages are more sensitive to salt stress compared to later stages [51]. High salt concentration leads to an increase in osmotic and ionic stresses. The majority of the ionic stress is caused by sodium toxicity which leads to ion imbalance, and hyperionic or hyper-osmotic stress disintegrates the overall metabolic activities leading the plant to death [51]. Salt stress leads to a reduction in leaf area, chlorophyll content and stomatal conductance which reduces productivity and juice quality. Plants have developed various self-defensive complex mechanisms, such as adaptation, accumulation and combinations of these, to protect themselves from the osmotic and ionic stresses caused by high salt concentration [52]. The adaptation mechanism mostly followed by plants to mitigate salt stress is a permanent genetic modification of the plant's structure and function [10]. In addition to the adaptation mechanism, a majority of the plants respond to intermittent stresses by modifying physiological or morphological changes by a process called acclimation, a reversible process that does not involve any permanent genetic changes [10].

### 3.3. Cold and Heat

Sugarcane production and productivity are severely affected by high and low-temperature stresses [53]. Cold stress severely affects the plants' metabolic processes and accumulates reactive oxygen species (ROS), which severely damages biomolecules such as DNA, proteins, carbohydrates, lipids, and pigments in the cell [54]. Cold stress causes shortening of internodes and leaves and stunting of canes, and also affects the ratooning ability [55]. Heat stress severely affects the germination and growth of seedlings despite having enough

water. High-temperature stress leads to drying and necrosis of leaf tips and margins, and reduces photosynthesis and sugar accumulation [56].

### 3.4. Waterlogging

Waterlogging mainly caused by excessive rainfall and flood affects sugarcane yield and juice quality [53,57,58]. Waterlogging during the formative stage causes a yield reduction of up to 45% [58]. Water stagnation for a prolonged period of time severely affects the crop metabolism, nutrient and water uptake, decreases sucrose content and increases ROS, glucose, total colloids, non-proteinous nitrogen and gums [57,58]. The symptoms of waterlogging are aerial root formation, reduction in leaf size and number, stunted growth, yellowing and wilting of canes, ultimately leading to death of the plants [53,58]. The aerial roots with aerenchymatic cells help in gas exchange under submerged conditions. Ethylene accumulation in root systems increases under waterlogging which aids in the production of aerial roots [58].

### 3.5. Heavy Metals

Macro and micronutrients are required to the plant for their normal growth and development [59]. Excess accumulation of these macro and micronutrients such as manganese (Mn), copper (Cu), nickel (Ni) and others in the soil are harmful to plants [59,60]. Recent activities, such as mining, exploitation of groundwater, excessive use of fertilizers and industrialization, have greatly contributed to the repeated accumulation of toxic heavy metals such as lead, cadmium, arsenic, mercury, chromium and many others in the soil [61,62]. Accumulation of these heavy metals adversely affect soil health and is also harmful to all biological organisms, from microbes to humans [62–64]. Heavy metal toxicity affects all essential plant biological processes, such as germination, respiration, photosynthesis, metabolic reactions and reproduction. Symptoms of heavy metal toxicity in plants are chlorosis, necrosis, stunted growth, senescence, yield reduction and death of plants [59,60].

## 4. Transgene-Free Genome Editing: Approaches and Applications

Improvements in genome editing technique enable the creation of transgene-free GE plants in a single generation which is highly beneficial for producing transgene-free mutants in vegetatively propagated crops such as sugarcane. These techniques eliminate the need for meiotic recombination-based selection of transgene-free plants in successive generations. The possible techniques that may be applied to produce transgene-free GE lines in sugarcane are illustrated in Figure 1.

### 4.1. Virus-Induced Genome Editing

Virus-induced genome editing (VIGE) is one of the promising approaches to develop transgene-free GE plants. Both DNA and RNA viruses can be used to deliver the gRNA and Cas endonucleases into plant systems. This can be performed in two ways: by delivering the gRNA into a plant system that already has Cas9 integrated into its genome, or by delivering both gRNA and Cas9 into the plant genome simultaneously [65]. The VIGE has been successfully employed in both model plants and economically important crops, such as tobacco, *Arabidopsis*, wheat, rice, maize, soybean, potato and tomato [65]. DNA viruses such as bean yellow dwarf virus (BeYDV), wheat dwarfing virus (WDV), cotton leaf crumple virus (CLCrV), and cabbage leaf curl virus (CaLCuV), and RNA viruses such as tobacco rattle virus (TRV), potato virus X (PVX), pea early browning virus (PEBV), tobacco mosaic virus (TMV), apple latent spherical virus (ALSV), beet necrotic yellow vein virus (BNYVV), tobacco etch virus (TEV), and foxtail mosaic virus (FoMV), have been successfully used for gRNA delivery into the plant system [66–84]. However, the serious limitation of using these viruses is their limited cargo-carrying capacity which makes it highly difficult to carry both gRNA and Cas9 in the same virus. These viruses can be efficiently used for introducing gRNA into plants that already have Cas9 in their genome. The negative-strand RNA viruses, namely barley yellow striate mosaic virus (BYSMV) and sonchus

yellow net virus (SYNV), have been used to produce targeted mutation in *N. benthamiana* by expressing both gRNA and Cas9 [85,86]. The large cargo-carrying capacity of these viruses has a potential application in delivering complete genome editing components into the host system. Since the size of Cas9 is large (~4.1 kb), it cannot be delivered together with gRNA in the plant system by most of the plant viruses due to their limited cargo-carrying capacity; this problem can be resolved by using small-size Cas variants, such as CasΦ and CasX, that enable them to deliver gRNA and Cas endonuclease together into the host plant [87,88]. The virus with genome editing components can be introduced into the host plant either by agroinfiltration or by mechanical method [65,82,89]. As the sugarcane is mainly propagated through setts, the following method can be effective for the VIGE approach in sugarcane. First, the viral vectors are agroinfiltrated into the growing canes and once the visible symptoms are expressed, the infected setts are propagated in the pot followed by meristem culture of the pot-grown setts to produce transgene and virus-free GE plants. Another approach is by subjecting canes to agroinfiltration, allowing them to set seeds and using the seeds as propagative material to develop transgene-free GE plants. Though the latter approach bypasses the need for tissue culture, it cannot be applied universally as sugarcane seed production is highly location and season-specific and the heterozygous nature produces more recombinants [90,91]. The main advantages of VIGE are the high gRNA expression due to the high proliferation rate of viruses, tissue culture-free development in the case of seed-propagated plants, less off-target effects and transgene-free plants [65,89]. Producing heritable mutations through the virus is difficult because the viruses cannot proliferate into the apical meristem or reproductive tissues. However, this problem can be overcome by fusing mobile RNA sequences, such as Flowering Locus T (FT), with gRNA to enable the virus to enter the shoot apical meristem portions to create heritable mutations without any residual transgenes [92].

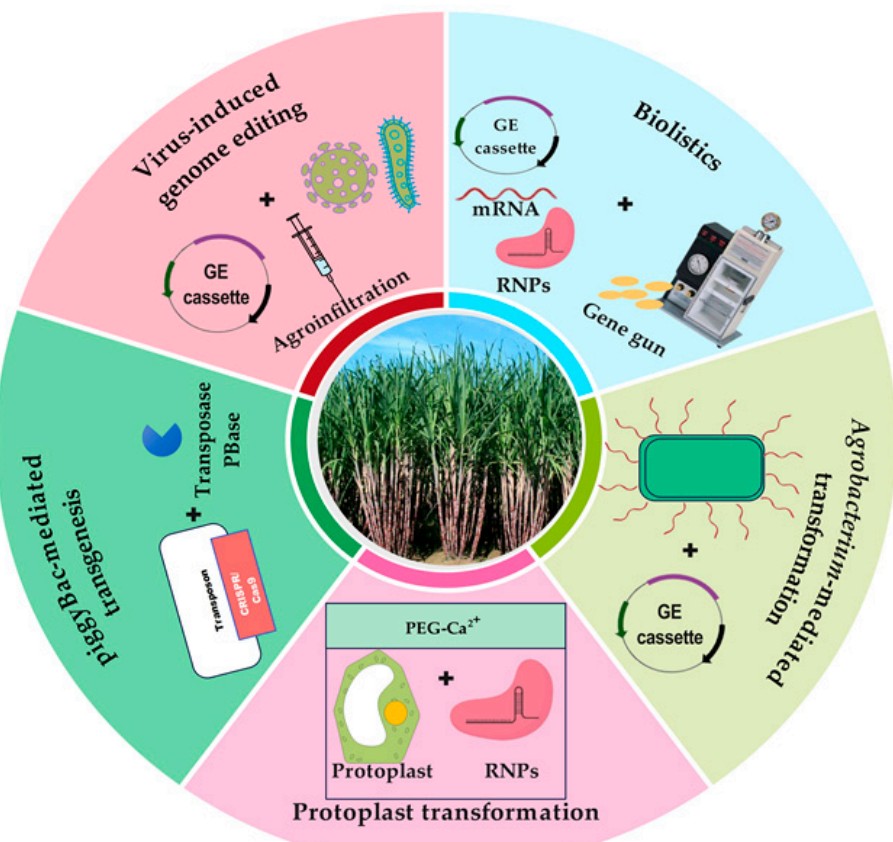

**Figure 1.** Approaches for producing transgene-free GE sugarcane lines.

### 4.2. Particle Bombardment/Biolistics

Particle bombardment/biolistics has been widely used for sugarcane genetic transformation along with *Agrobacterium*-mediated transformation [11,93]. Biolistics can deliver a variety of molecules, such as DNA, RNA, plasmid, and proteins, directly into the cells [94]. Delivering the plasmids containing genome editing cassette through biolistics randomly integrates the genome editing system into the plant genome, and the presence of foreign genes from plasmids into the plant genome leads the GE plants to fall under the category of GMOs. This problem can be circumvented by delivering ribonucleoproteins (RNPs) consisting of in vitro assembled gRNA and Cas9 through particle bombardment. This method uses Cas9 or cpf1 endonuclease that interacts with gRNA without any additional auxiliary factors [95]. The assembled genome editing reagents, such as Cas endonuclease and gRNA molecule, are active only for a short period of time and become degraded inside the cell. The RNP is the mixture of Cas9 protein and gRNA molecules in definite preparation along with other components such as a buffer [96]. The biolistics delivery of RNPs has been successfully used for genome editing in crops, such as rice, wheat and maize [96–99]. In sugarcane, embryogenic callus is widely used for particle bombardment and many successful transgenic events were developed using biolistics [11,93]. Biolistics was successfully used for delivering the CRISPR/Cas9 genome editing cassette into sugarcane callus by Oz, et al. [100] and Eid, et al. [101]. Oz, et al. [100] edited the alleles of the *acetolactate synthase* (*ALS*) gene to produce herbicide resistance. Upon spraying the herbicide nicosulfuron, the wild types died whereas the GE lines survived. Eid, et al. [101] edited the multiple alleles of the *magnesium chelatase* gene, which is essential for chlorophyll biosynthesis, and the edited plants were yellow in color whereas the wild-type plants showed normal green color. These studies highlighted the potential use of biolistics in CRISPR/Cas-mediated genome editing to produce targeted mutations in sugarcane.

### 4.3. Agrobacterium-Mediated Transformation

*Agrobacterium*-mediated transformation has been used for genetic transformation in sugarcane to produce transgenic events for various purposes [11,93]. Transient expression of gRNAs and Cas9 using *Agrobacterium* eliminates the integration of foreign genes into the plant system. This technique has been used to produce transgene-free GE plants in tobacco, tomato and potato [102–106]. Chen, et al. [103] targeted the *PDS* gene in tobacco by this method and generated transgene-free GE tobacco without using any selection media. Veillet, et al. [104] delivered Cytosine base editors by *Agrobacterium*-mediated transformation to edit the *acetolactate synthase* (*ALS*) gene to produce chlorsulfuron-resistant potato and tomato plants and obtained 10% and 12.9% of transgene-free potato and tomato plants, respectively [104]. The genome edited plants produced by this method contain a mixture of transgenic and non-transgenic individuals [103–106]. Leaf rolls and callus are the main explants used for *Agrobacterium*-mediated transformation which needs in vitro regeneration for its development. Mayavan, et al. [107], Mayavan, et al. [108] successfully demonstrated the in planta transformation using sugarcane seeds and setts through *Agrobacterium*-mediated transformation. This method bypasses the need for time-consuming and resource-intensive tissue culture facilities to produce transformed lines [107,108]. This opens a new window to exploit the possibilities of producing transgene-free GE plants using in planta transformation method.

### 4.4. Protoplast Transformation

The RNPs are delivered into protoplast through lipofection or PEG-Ca$^{2+}$ transfection. The lipofection method has been used in crops such as cabbage, Chinese cabbage and tobacco [109,110]. PEG-Ca$^{2+}$ transfection was the method mostly used for delivering the genome editing RPNs into the protoplasts and was used to produce transgene-free GE plants in model plant system and crop plants, such as *Arabidopsis*, tobacco, lettuce, maize, rice, banana, canola, wild tomato and potato [111–116]. In the case of sugarcane, electroporation and PEG-mediated protoplast transformation were successful in producing

transgenic events, but transformation and regeneration efficiency was low [117–120]. Wang, et al. [121] optimized the protoplast isolation and transformation through PEG, which had a protoplast yield of approximately $1.26 \times 10^7$ per gram of leaf and achieved a transformation efficiency of 80.19% in *Saccharum spontaneum* [121]. Though efficient protoplast isolation and transformation methods have been developed, the regeneration of whole plants from the sugarcane protoplast is genotype specific and still a promising issue that needs to be addressed. By developing an efficient protoplast regeneration technique, the protoplast transformation could potentially be exploited for producing transgene-free GE sugarcane.

### 4.5. piggyBac-Mediated Transgenesis

Nishizawa-Yokoi and Toki [122] used *piggyBac* transposon to remove the randomly integrated CRISPR/Cas9 genome editing cassette from the rice genome. This system enables editing of the target site by temporary expression of CRISPR/Cas9 followed by the elimination of transgenes from the rice genome by *piggybac* transposase (PBase) [122]. This system successfully produced transgene-free genome-edits in rice and this technology opens a new avenue for eliminating the transgenes in vegetatively propagated crops such as sugarcane, where eliminating transgenes through recombination and selection is highly difficult and time-consuming [122,123].

### 4.6. Grafting and Mobile RNA-Mediated Genome Editing

A novel technique using grafting for transgene-free GE plants was demonstrated by Yang, et al. [124] in *Arabidopsis thaliana* and *Brassica rapa*. The fusion of tRNA-like sequences (TLS) has produced a mobile version of CRISPR/Cas9 RNA which enables the root-to-shoot movement from root stock to scion. *Agrobacterium*-mediated transformation was used to deliver *Cas9*-TLS mRNA and gRNA-TLS fusions into *Arabidopsis thaliana* which was used as root stock for both *Arabidopsis* scion and *B. rapa* scion [124]. The hypocotyl grafting was done between transformed *A. thaliana* root stock containing the genome editing cassette and wild type scions of *Arabidopsis* and *B. rapa*. The fusion product moved from root stock to scion with the aid of TLS and created edits in the genome of scions, while the edited scions produced seeds with transgene-free targeted mutation [124]. However, this method needs to be established in sugarcane because the monocots are devoid of vascular cambium and have scattered vascular bundles. Vascular cambium is required for successful graft establishment [125].

## 5. Screening of Transgene-Free GE Lines

Plasmid-based delivery of CRISPR/Cas genome editing cassette produces a mixture of transgenic and non-transgenic individuals. The techniques such as the transgene killer CRISPR system (TKC) and visual marker system help in the easy identification and selection of transgene-free GE lines generated by plasmid-based delivery methods.

TKC uses suicidal genes that kill the male and female gametophyte-containing transgenes and allows only the seed produced from the fusion of transgene-free pollen and embryo [126,127]. Two cassettes, one containing the *barnase* gene and the other with *CMS2* were added into the CRISPR/Cas9 construct. The *barnase* gene under the control of rice *REG2* promoter kills the embryo that contains transgenes and the *CMS2* gene under the control of *CaMV 35S* promoter disrupts mitochondrial functions during male gametophyte development and causes male sterility [126,127]. Yang, et al. [128] demonstrated the potential of the TKC system along with multiplexing to edit multiple gene families in plants and produced multiple transgene-free null mutant combinations. This system can be potentially used for seed propagating plants, but for vegetatively propagated crops such as sugarcane, where flowering and seed set is highly difficult, this system is a big challenge to implement to produce transgene-free edits.

Transgene-free GE seeds/plants can be easily identified visually by utilizing fluorescent markers and pigments [123]. Introducing fluorescent marker proteins, such as mCherry, along with genome editing cassette produces bright color in transgenic seeds

which enables easy differentiation of transgene-free GE seeds [129,130]. By using different marker proteins, such as DsRED, or green fluorescent Proteins (GFPs,) the same strategy was used to select transgene-free seeds in rice, *Arabidopsis*, tomato and maize [131–133]. The other approach to visually select transgene-free GE plants is by introducing genes such as the *PAP1* gene of anthocyanin biosynthesis pathway or artificially synthesized *RUBY* reporter gene for betalain synthesis pathway in the genome editing cassette that produces purple and vivid red color, respectively in the transgenic calli and plants [134,135]. In sugarcane's case, delivering the genome editing cassette through *Agrobacterium*-mediated transformation or biolistics, followed by selecting the transgene-free plants through a visual marker system, particularly pigment-based selection during callus stage, will eliminate the plants with integrated transgene at an early stage and reduce the cost and workload involved in analyzing GE lines for targeted mutation.

The transgene-free lines developed through RNP-based delivery systems can be analyzed for targeted mutation using a variety of PCR-based approaches, such as PCR-RFLP, T7 endonuclease 1(T7E1), semi-nested PCR, Droplet Digital PCR and Cas9/sgRNA-assisted reverse PCR (CARP), that can be used for preliminary screening of mutants [136–140]. Among these techniques, PCR-RFLP and T7E1 are mostly used techniques, and in PCR-RFLP, the PCR products are subjected to restriction digestion [139,140]. Restriction enzyme cleaves the target region in wild types whereas mutants remain unaffected. Site-specific mutation created by CRSIPR/Cas reagents leads to loss of restriction site in the genome-edited plants [139,140]. The other reliable technique to screen mutated lines is T7 endonuclease 1 (T7E1) assay [141–144]. During PCR amplification of mutants, the base changes at the target site cause heteroduplex formation; upon treating the amplified products with T7 endonuclease 1, the heteroduplex portion in mutants will be cleaved and the wild type remains unaffected [141–144]. These assays enable the preliminary screening of mutated lines and limits the number of individuals for further sequencing and analysis (Figure 2).

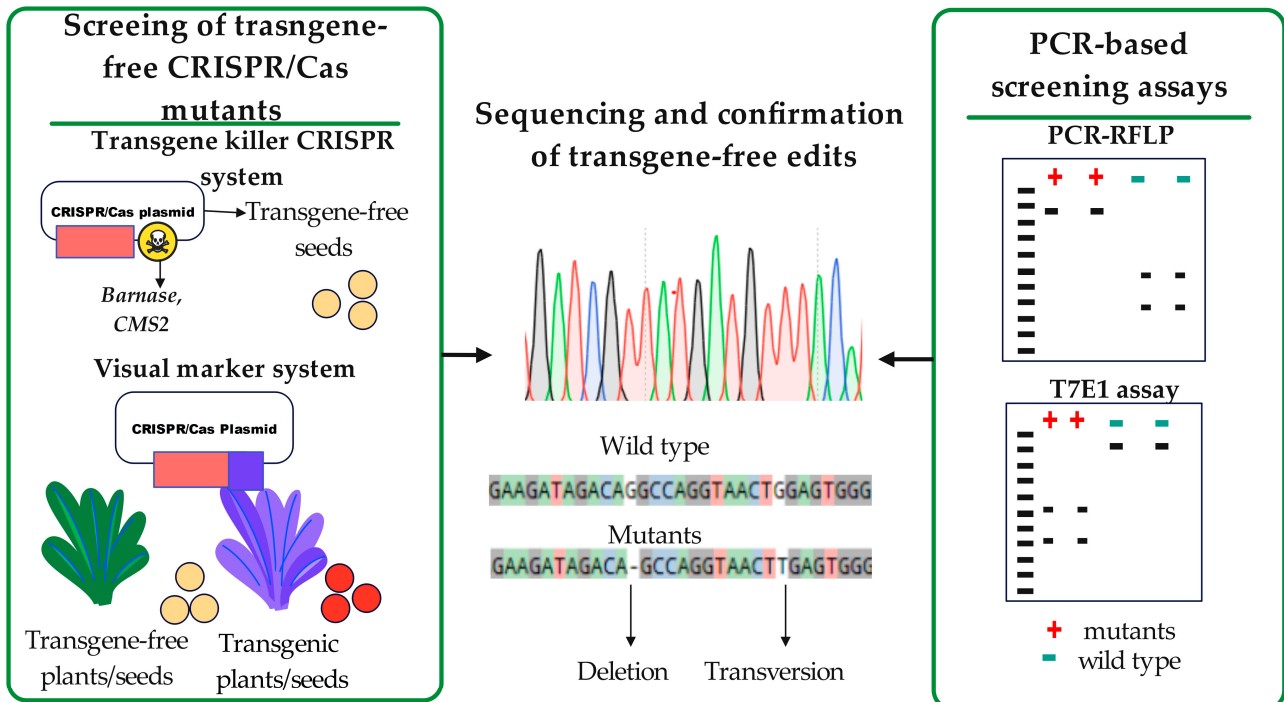

**Figure 2.** Strategies for screening and analyzing the transgene-free GE lines. TKC and visual marker system are used for selecting transgene-free GE plants/seeds developed through plasmid-based delivery methods. The plants developed through RNP-based delivery systems are subjected to PCR-RFLP and T7E1 assays to identify mutated individuals. Finally, mutants identified in these methods are further sequenced and analyzed for type of mutation in the targeted region.

## 6. Candidate Genes for Genome Editing in Sugarcane for Biotic and Abiotic Stress—Lessons from Crop Plants

In sugarcane, a number of candidate genes were identified for biotic and abiotic stresses (Table 1) and targeting these genes through the CRISPR/Cas system may produce multiple stress-resistant/tolerant cultivars. Other than the genes listed in Table 1, many putative candidate genes and genomic loci have been identified in sugarcane and its wild species for various biotic and abiotic stresses [55,145–149].

**Table 1.** Candidate genes for biotic and abiotic stresses in Sugarcane.

| Stress | Candidate Genes | References |
|---|---|---|
| Drought | *SoACLA-1* | [150] |
| Drought | *SoP5CS* | [151] |
| Drought | miRNAs | [152,153] |
| Drought | *ScLoX, Dehydrin. Dirigent-jacalin* | [154] |
| Salinity | miRNAs | [155] |
| Salinity | *ShPHT* | [156] |
| Drought and salinity | *SodERF3* | [157] |
| Drought and salinity | *SoMYB18* | [158] |
| Drought and salinity | *Scdr2* | [159] |
| Drought, salinity and oxidative | *Scdr1* | [160] |
| Drought, salinity and oxidative | *ScDir* | [161,162] |
| Drought, salinity and oxidative | *ScAPX6* | [163] |
| Salinity and low temperature | *SspNIP2* | [164] |
| Drought, salinity and cold | *ShGPCR1* | [165] |
| Whip smut | *ScCAT1* | [166] |
| Rust | *Bru1* | [167] |

Genome editing has been successfully used to create GE lines resistant/tolerant to various biotic and abiotic stresses in many crop plants [168–170]. Structural genes, regulatory genes, and *cis*-regulatory elements (CREs) of regulatory and structural genes are the potential targets for CRISPR/Cas system to produce abiotic stress resistant/tolerance plants [168]. In the case of biotic stresses, resistance can be induced either by manipulating the susceptible factors of the host that favors pathogen establishment and spread or by targeting the pathogen factors. Table 2 summarizes some of the recent reports on the use of CRISPR/Cas genome editing in plants for various abiotic and biotic stresses. Most of the published work on transgene-free genome editing was done in model plants, such as *Arabidopsis thaliana* and *Nicotiana bethamiana*, by targeting the genes that enable easy visual identification such as *PDS*. Only very few reports have been published on transgene-free genome editing for biotic and abiotic stresses in crop plants (Table 3).

**Table 2.** CRISPR/Cas mediated genome editing for various biotic and abiotic stresses in crops.

| Stress | Crop | Target Gene | References |
|---|---|---|---|
| Drought | Rice | *OsERA1* | [171] |
| | | *OsDST* | [172] |
| | | *OsPYL9* | [173] |
| | | *OsWRKY5* | [174] |
| | Tomato | *SlGID1* | [175] |
| | | *SlMAPK3* | [176] |
| | Wheat | *TaSal1* | [177] |
| | | *Sal1* | [178] |
| | Soybean | *GmHdz4* | [179] |
| | Chickpea | *4CL* and *RVE7* | [180] |
| Salinity | Rice | *OsbHLH024* | [181] |
| | | *OsRR22* | [182] |
| | | *OsmiR535* | [183] |
| | Tomato | *SlARF4* | [184] |
| | | *SlHyPRP1* | [185] |
| | Soybean | *GmAITR* | [186] |
| Heavy metals | Rice | *OsNRAMP5* | [187] |
| | | *OsNRAMP1* | [188] |
| | | *OsARM1* | [189] |
| Fungus | | | |
| Rice Blast | Rice | *OsERF922* | [190] |
| | Rice | *OsSEC3A* | [191] |
| | Rice | *Bsr-d1*, *Pi21* and *ERF922* | [192] |
| Powdery mildew | Tomato | *SlML* | [193] |
| | Tomato | *SlPMR4* | [194] |
| | Wheat | *TaMLO-A1* | [195] |
| | Grapevine | *VvMOL3* | [196] |
| Stem rot | Rapeseed | *BnWRKY70* | [197] |
| Black pod | Cacao | *TcNPR3* | [198] |
| Ergot | Rye | *pyr4* and *TrpE* | [199] |
| Bacteria | | | |
| Bacterial leaf blight | Rice | *Pi21* or *ERF922* | [192] |
| | | *OsSWEET14* | [200] |
| Bacterial leaf spot | Tomato | *SlJAZ2* | [201] |
| | | *Sldmr6-1* | [202] |
| Banana Xanthomonas wilt | *Musa balbisiana* | *LRR, WAK2, WAK5, Vicilin, RPM1, PR1, NPR1* | [203] |
| | Banana | *MusaDMR6* | [204] |
| Bacterial speck | Tomato | *SLDMR6-1* | [202] |

**Table 2.** *Cont.*

| Stress | Crop | Target Gene | References |
|---|---|---|---|
| | | Virus | |
| Rice tungro spherical virus | Rice | *eIF4G* | [205,206] |
| Rice black streaked dwarf virus | Rice | *eIF4G* | [207] |
| Rice black-streaked dwarf virus | Maize | *ZmGDIα* | [208] |
| Potato virus Y | Potato | *eIF4E1* | [209] |
| | | *eIF4E* | [210] |
| Tomato brown rugose fruit virus | Tomato | *TOM1* | [211] |
| Tomato yellow leaf curl virus | Tomato | *IR, CP* of virus | [212] |
| Soybean mosaic virus | Soybean | *GmF3H1, GmF3H2* and *GmFNSII-1* | [213] |
| Cucumber vein yellowing virus, Zucchini yellow mosaic virus; Papaya ring spot mosaic virus- | Cucumber | *eIF4G* | [214] |
| Banana streak virus | Banana | *ORF1, ORF2,ORF3* of virus | [203] |

**Table 3.** Transgene-free genome editing in crops.

| Method | Stress | Crop | Target Gene | References |
|---|---|---|---|---|
| Protoplast transformation | Drought | Chickpea | *4CL* and *RVE7* | [180] |
| *Agrobacterium*-mediated transformation | Bacterial leaf blight | Rice | *Xa13* | [215] |
| *Agrobacterium*-mediated transformation | Powdery mildew | Tomato | *SlMlo1* | [193] |

## 7. Regulatory Issues in Commercializing GE Plants

The regulatory approvals for commercial utilization of GE products is location specific as it varies from country to country [216]. Generally, GE plants are regulated in two ways, namely product-based regulation and process-based regulation [217,218]. In the case of product-based regulation, the health and environmental risks posed by GE products are assessed by the final product of the technique and not with the methodology associated with it; whereas the process-based regulation, considers the techniques used for the development of GE crops for risk assessment. The process-based regulation restricts the commercialization of GE crops created with the aid of foreign genes even though the end products are devoid of any foreign genes [217,218]. This severely affects the commercial exploitation of GE crops and future developments in genome editing. Countries such as USA, Canada, Argentina, Australia, Russia, and Chile follow product-based regulation, whereas the European Union follows the process-based regulation [217,219]. Based on double-stranded breaks (DSBs) produced by site-directed nucleases (SDNs), the GE products are categorized into three classes, namely SDN1, SDN2 and SDN3 [219,220]. The SDN1 system repairs the DSBs through NHEJ without inserting any foreign genes and it is similar to that of natural mutations [219,221]. The SDN2 system repairs DSBs using a donor template with a few base changes and the resulting product is similar to that of SDN1 without any transgenes [219,222]. The SDN3 system inserts a gene or large DNA sequences at the DSBs, and the product is either transgenic or cisgenic based on the donor sequence [219,220,223]. SDN1 and SDN2 products are exempted from regulation in many countries, such as the USA, Canada, India, China, Russia, Switzerland, Japan, Philippines, Australia, Kenya, Nigeria, Argentina, Chile, Ecuador, Brazil, Columbia, Paraguay, Honduras and Guatemala [224]. In the European Union and New Zealand, GE crops are categorized under GMOs and in countries such as Indonesia, Bangladesh, Uruguay and Norway, commercialization of GE crops is under consideration [224]. Globally, unprecedented changes in climate and increasing demand for food highlight the need for use of potential technologies such as

genome editing in the crop improvement process to meet all kinds of demands. In order to meet global food demands, the world nations that are restricting the commercialization of GE crops should consider the use of SDN1 and SDN2 products for commercial cultivation.

## 8. Challenges in Transgene-Free Genome Editing in Sugarcane

Transgene-free genome editing opens a new window for commercializing the GE lines of sugarcane, since eliminating transgenes, such as Cas endonuclease, promoters, marker genes and other components, from genome editing cassette in the host plant genome through genetic recombination is highly difficult. Although transgene-free genome editing techniques have numerous advantages over transgenics, many factors, such as long duration, complex genome with varying chromosome numbers ranging from 80–130, high polyploid nature with multiple alleles of a single gene, heterozygosity, and very high genome size, pose difficulties in establishing an efficient genome editing platform in sugarcane [225–227]. The availability of well-annotated whole genome sequence information enables the use of gRNA design tools, such as CHOP-CHOPv3, CRISPR-P 2.0 and many others, to select the gRNAs targeting the selective regions in the genome with good editing efficiency and less off-target site binding [228,229]. Unfortunately, such genome information is lacking in sugarcane and even a 1 or 2 bp mismatch between gRNA sequences and the host genome's target region prevents the expression of the genome editing system inside the host plant. The release of the draft of the whole genome sequence of sugarcane hybrid cultivar SP80-3280 in 2017 [230], the monoploid reference genome of sugarcane cultivar R570 [231] and the allele-defined genome of a haploid *Saccharum spontaneum* AP85-441 [232] in 2018 provides the opportunities to exploit CRISPR/Cas technology in sugarcane for various traits. Even though the reference genome sequences are available, it should not be directly used as reference for gRNA designing. It is highly recommended to clone and analyze the sequence of the target gene from the cultivar chosen for genome editing prior to gRNA designing. This process helps avoid the nucleotide mismatches between gRNA and the target region and ensures the correct binding of gRNA in the target site. The lower transformation efficiency of sugarcane, when compared to other crops, calls for the development of efficient transformation techniques in sugarcane [226,227,233]. Despite there being many advantages in transgene-free techniques, the low editing efficiency and the lack of selection pressure for the identification of transgene-free GE plants at an early stage not only make the process more complicated, but also more expensive [95]. However, these limitations can be circumvented by using visual markers, such as pigments and fluorescent protein markers, for the identification and elimination of the transgene-containing sugarcane plants at early stages and limiting the number of plants for further sequencing and analysis [123]. The VIGE, protoplast transformation of RNPs and many other techniques show promising results in producing many transgene-free GE crops, but their use in sugarcane is yet to be explored [95,123,234].

## 9. Conclusions and Future Prospects

Crop improvement has seen a dramatic shift from traditional breeding methods, where the selection is purely based on the phenotype, to genome-based selection of an individual or the population. Phenotype-based methods are highly laborious and time-consuming, which requires approximately 12–15 years to develop a variety based on the duration and reproductive nature of the crop plants [225,235]. Though transgenic technology has great potential to create plants resistant/tolerant to various biotic and abiotic stresses, commercial success in food crops is very limited across the world due to the various regulatory concerns surrounding the cultivation of transgenic food crops. The advent of genome editing technologies, such as ZFNs, TALENS and CRISPR/Cas, helps to produce site-specific mutations in the plant genome within a very short span of time. Conventional plasmid-based delivery of CRISPR/Cas genome editing cassette has been successfully employed in various crop plants and has produced promising results in the creation of novel alleles, new traits/products. Nevertheless, the integration of transgenes

from plasmids into the plant genome places the GE lines under GMOs [216,217]. Crop plants that reproduce through seeds such as rice, maize, tomato and many others, have the advantage of meiotic recombination to eliminate transgenes from their genomes. In the case of vegetatively propagated crops, especially sugarcane, the high heterozygosity along with the complex ploidy level pose difficulties in eliminating the transgenes through meiotic recombination. Moreover, exact genetic reconstitution is highly difficult due to the complex nature of the genome. PCR-based assays, visual marker system and TKC enable the preliminary screening of transgene-free genome edits. Employing transgene-free genome editing approaches, such as particle bombardment, *Agrobacteruim*-mediated transformation, VIGE, protoplast transformation and *piggyBac*-mediated transgenesis, along with advanced screening approaches, likely produce transgene-free GE lines resistant/tolerant to biotic and abiotic stresses.

**Author Contributions:** Conceptualization, S.S.K. and C.A.; methodology, S.S.K., S.R.H.C., C.A.; validation, R.V. (Ramanathan Valarmathi), K.L., M.R. and R.V. (Rasappa Viswanathan); writing—original draft preparation, S.S.K., S.R.H.C., M.R. and C.A.; writing—review and editing, R.M., M.R., S.S.K., R.V. (Ramanathan Valarmathi), G.H., P.T.P. and C.A. All authors have read and agreed to the published version of the manuscript.

**Funding:** Science and Engineering Research Board (SERB), Department of Science and Technology (DST)—Core Research Grant (F. No. CRG/2020/003761).

**Acknowledgments:** The authors thank the Director of the ICAR-Sugarcane Breeding Institute for providing facilities and support.

**Conflicts of Interest:** The authors declare no conflict of interest.

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
