# Peer review of "Transgene-Free Genome Editing for Biotic and Abiotic Stress Resistance in Sugarcane: Prospects and Challenges"

_agronomy, doi:10.3390/agronomy13041000_

Round 1

Reviewer 1 Report

Dear authors,

Sugarcane is one of the most valuable food and industrial crops. However, its production is severely affected by many biotic and abiotic stresses. In response to this problem, the gene-free genome editing technique is a good solution. The authors summarized the biotic and abiotic stresses encountered in sugarcane production, and introduced the application of exogenous-free genome editing in sugarcane, which provides a reference for sugarcane production. Overall, the manuscript is well written and fluent. However, some issues need to be solved before publication.

Major Comment:

1.      The abstract is full of background description rather than describe the main content of this article.

2.      In the ‘abiotic stress’ section, why has only drought stress been discussed but waterlogging stress not?

3.      ‘4. DNA-free genome editing: Approaches and Applications’, has been divided into as many as 8 sections, which is unnecessary in my opinion. ‘4.6. Transgene killer system’ and ‘4.7. The two sections of the Visual marker system’ are not about the transformation technology, instead they are related to the screening of genetically modified plants. Besides, some of these transformation technologies are not used in sugarcane currently (4.5. piggyBac-mediated transgenesis, 4.8. Grafting and mobile RNA mediated genome editing). Therefore, this part should be modified accordingly.

4.      The wording in ‘6. Regulatory issues in commercializing genome-edited plants’, ‘7. Challenges in transgene-free genome editing in sugarcane’ and ‘8. Conclusion and future prospects’ is lengthy and should be streamlined or consolidated.

Minor Comments:

1.      Line 193, Sugarcane (Saccharum officinarum L.) should be placed where ‘sugarcane’ is introduced for the first time in the article.

2.      In Fig1, ‘Transgene killer system’ and ‘CRISPR’ should be arranged in the same row, and the order of the parts should preferably follow the order in which the article is discussed.

3.      For ‘(SlJAZ2’ appears in Table2 ‘Bacteria’, please remove the extra parentheses, and adjust the width of ‘Crop’ column to ensure that ‘Musa balbi-Siana’ is in one row.

4.      line 107, Why has ‘Pokkah boeng’ been written in italics? Other diseases are not in italics.

5.      In this article, the reduction due to disease/pest infestation has been mentioned several times and the detailed percentage of yield reduction has been listed, but the source of the data has not been indicated, for example, line 150 and 156. Is there any reference for this part?

Reviewer 2 Report

Comments to the Author:

In the review of “Transgene-free genome editing for biotic and abiotic stress resistance in sugarcane: Prospects and Challenges”. The author introduced the method and application of DNA-free genome editing, and emphasized its huge application potential in sugarcane, especially in response to biological and abiotic stresses. The review is more interesting, in my opinion, just a few changes are required.

Specific recommendations:

1. The second part of the manuscript introduces the major biological and abiotic stresses suffered by sugarcane. The content of this part is not the focus of this paper, and it is a little boring in language. The author can consider making a table to explain the content more succinctly and intuitively.

2. Some formats in the manuscript need to be adjusted, for example, the which in line 101 should not be italicized.

3. Grammatically speaking, "upto" is incorrect when a part of a sentence. Line 102 and line 104, "upto" should be "up to".

4. Line 181, Superoxide radials should be "O2-" instead of "O-2".

5. Some nouns in the article are incomplete, affecting readers' understanding of the content. Line 264, the small Cas variant "Cas " should be "Cas X". Table2 "(SlJAZ2" should be "SlJAZ2". Table 2. ,"OsWRKY5" should be "OsWRKY5".

6. Note that some of the same words should use the same format in the full text, For example, line 311 "Agrobacterium-mediated" line 318 "Agrobacterium-mediated".

Reviewer 3 Report

Here are my comments to the manuscript: agronomy-2290690-peer-review-v1, which has been submitted to Agronomy, (Transgene-free genome editing for biotic and abiotic stress resistance in sugarcane: Prospects and Challenges by Sakthivel et al.) The authors are writing a review article highlights information, that how transgene-free genome editing techniques will be utilized to improve sugarcane crop. The manuscript was conducted using some previous published work in the field. The manuscript contains information that would be of interest to the readers. But the work as presented is not enough and need more, details, deep discussion, and analyses of the previous published work. Not only mentioning and listing results of the previous works, but more deep discussion and analyses. The authors are just on the surface of the discussion in some Sections.

Comments and Suggestions for Authors

1.      English language and style are fine/minor spell check required.

2.      The bulk of the abstract discusses general information’s related sugarcane crop. I feel it should summarize more research progress in the review and outline the content of the manuscript. So it is better to revise the abstract following the clue of the main text of the manuscript

3.      The significance of the recent manuscript is missing in introduction and the objectives are not clear, why authors has chosen this topic?

4.      In the review article, I am missing a summary of each paragraph especially in Section 2 & 3 and their sub sections. Please add more in-depth details to each section.

5.      The graphical figure is good but missing more explanation and details.

6.      The authors divided their work into main points; Biotic and Abiotic stresses, DNA-free genome editing, and, that ok, but more deep discussion is missing in each section.

7.      Remove sub heading especially in section 2 & 3 and keep the main headings.

8.      Please add more information to the table 1 as the information’s are not enough.

9.      Add a schematic representation indicating molecular mechanism of stress tolerance in sugarcane such signaling cascades, Interaction networking etc.

10.   If the authors could provide some insights and perspectives of their own. What is the main problem still facing the research community? What are the urgent tasks in the future? This could be better to add in Section#7.

Round 2

Reviewer 1 Report

The paper can be accepted without any further changes.

Reviewer 3 Report

Thank you for your revision.